# An Investigation of Learning Needs in the Mining Industry

Sergio Miranda *, Antonio Marzano and Rosa Vegliante

Department of Human, Philosophical and Educational Sciences, University of Salerno, 84084 Fisciano, Italy; amarzano@unisa.it (A.M.)
* Correspondence: semiranda@unisa.it

**Abstract:** Mining operations are risky and often dangerous, with a high potential for accidents. Many of these accidents can be prevented by implementing safety measures. It is essential that mining companies take these measures seriously to protect the safety and wellbeing of their workers and ensure the sustainability of the industry. Among these measures, those related to training are addressed in this paper in relation to the ERASMUS+ project entitled DigiRescueMe, which aims at developing courses to increase the knowledge and level of awareness of miners, rescue members, and mining engineers and, consequently, reduce the death rate in mine accidents. For this goal, semi-structured interviews and surveys were implemented, and the collected data were analyzed. The mining industry is a wide domain connected to other sectors like universities, vocational schools, rescue centers, and agencies. For this reason, the investigations carried out herein engaged people from all these sectors to identify firstly the themes, secondly, the topics, and finally the knowledge levels corresponding to those themes and topics in order to determine the learning needs and translate them into requirements for the courses that will be developed during the project activities.

**Keywords:** learning needs; learning needs analysis; mining industry; learning; training

## 1. Introduction

Mining is a critical industry that supplies essential resources for society, including coal, metals, and minerals. Mining operations are inherently risky and often dangerous, with a high potential for accidents. These accidents can lead to serious injuries, fatalities, and economic losses. There are many distinct types of accidents that can occur in mining, but some of the most common ones include cave-ins, explosions, fires, haulage accidents, and falls.

Cave-in accidents are caused by the collapse of underground mines, which can result from weak rock strata, over-extraction, or natural disasters like earthquakes. Explosions can occur due to the build-up of gases like methane or coal dust, which can ignite. Fires can be caused by faulty electrical equipment, friction, or the spontaneous combustion of coal. Haulage accidents occur when vehicles and equipment used in mining collide with each other, roll over, or overturn. Falls occur when miners fall from heights or when materials or equipment fall on them.

Mining accidents can be caused by several factors. The first is poor safety management, which means a lack of safety training for workers. Poorly maintained equipment or the use of faulty equipment can result in accidents. Human error can occur when workers are fatigued, inexperienced, or do not follow safety protocols. Also, natural disasters like earthquakes, floods, and landslides can cause mining accidents. Finally, ignoring safety regulations and guidelines can result in accidents. Several measures can be taken to prevent accidents in mining. Some of these measures are related to engineering, maintenance, equipment, and systems. Mining accidents can have profound consequences, including the loss of life, injuries, and economic losses. Many of these accidents can be prevented by implementing safety measures such as safety training, equipment maintenance, adherence to safety regulations, and improved ventilation. It is essential that mining companies

take these measures seriously to protect the safety and wellbeing of their workers and to ensure the sustainability of the industry. Among these measures, those related to training were addressed in this study, and an initial analysis was conducted to find the real learning needs.

A review of fatal incidents in the Western Australian mining industry from 2000 to 2012 [1] analyzed data from the Western Australian Department of Mines and Petroleum and found that there were 52 fatal incidents during that time period. The incidents were categorized into six diverse types: vehicle-related, fall of ground, machinery-related, drowning, explosives-related, and other. The authors also found that most of the incidents occurred in the gold mining sector, and that the most common contributing factors were human error, inadequate training, and inadequate supervision.

In an analysis of fatalities and injuries that occurred in the United States mining industry between 2003 and 2014, with a focus on incidents involving mining equipment [2], the causes and contributing factors of these incidents were identified. The authors found that incidents involving mining equipment were the second most common cause of fatalities in the mining industry, accounting for 26% of all fatalities. They also found that the most common types of equipment involved in fatal incidents were haul trucks, followed by loaders and bulldozers. The authors identified several key factors that contributed to incidents involving mining equipment, including inadequate maintenance and inspection, operator error, and design and engineering deficiencies. They also named several strategies for improving safety in the mining industry, including the use of advanced technologies, improved training and education for equipment operators, and the development of more robust safety regulations and guidelines.

In a review of accident analysis and safety improvements in underground coal mines in China [3], the authors examined data on coal mine accidents and safety measures from various sources, including government reports, academic articles, and industry publications. They found that, despite considerable progress in reducing coal mine accidents in China, underground coal mining remains a hazardous activity, with a high number of fatalities and injuries. This paper discussed the main causes of coal mine accidents in China, which included inadequate safety management systems, a lack of training, and the inadequate maintenance of equipment. The authors also identified some key safety improvement measures, such as the use of modern mining technologies, enhanced safety regulations, the establishment of safety management systems, and better safety training for workers to promote a safety culture in the mining industry.

The learning needs for safety competence in the mining industry were explored in Ding, Yuan and Liu [4], and a framework for identifying and addressing these needs was proposed.

The learning and development needs of future mining workforces in light of technological advancements in the industry were examined in Williams and Huddlestone-Holmes [5], in order to argue that mining companies need to invest in upskilling their workers and fostering a culture of continuous learning to keep pace with technological change.

Moreover, a model for an industry-driven learning ecosystem in the mining industry that can support the upskilling of the workforce was proposed in Simms and Dixon [6] to help to address the skill gaps that currently exist in the industry and enable mining companies to adapt to changing technology and market conditions.

Attention has been paid to training and learning, but although many research activities have been conducted and several papers have been published in recent years on the topic of learning needs in the mining industry, there is still a lot of confusion as to both the methodological approaches to follow and the issues on which it is appropriate to intervene.

The focus of work pedagogy is promoting learning and development in the workplace by referring to the theories and assumptions related to knowledge and learning that underlie educational practices, shaping the ways in which workers engage in learning activities and acquire new knowledge and skills. One important dimension of the epistemological framework of work pedagogy is the practical–experiential dimension [7]. This dimen-

sion emphasizes the importance of direct learning experiences, where workers engage in activities that directly apply to their work tasks and responsibilities. This approach is grounded in the belief that learning is most effective when it is contextualized within a specific work environment and is closely aligned with the goals and needs of the organization [8]. Another important aspect is considering not only the technical skills and knowledge needed for work tasks, but also the social, emotional, and psychological dimensions of learning and development. The aim is to create a positive work environment that supports workers' wellbeing and promotes their engagement and motivation [9]. These principles are fundamental for the mining industry, where the conditions of work are hard and the wellbeing of workers is one of the goals of any initiative, especially if its aims are learning and training. These were the main reasons behind the learning needs assessment conducted in this research.

## 2. Learning Needs Analysis

Regardless of the learning objectives, which, according to their nature, can be based in the cognitive (mental), affective (attitude), and psychomotor (physical) domains [10], learning needs analysis is a domain in which scientists and pedagogists have conducted studies and research and published key ideas and perspectives. In the field of early childhood education, Maria Montessori is a pioneering figure who emphasized the importance of seeing and understanding the individual needs and interests of each child [11]. She believed that children have an innate desire to learn, and that the role of the educator is to create an environment that supports this natural curiosity. Montessori encouraged teachers to use observation and assessment tools to find each child's unique learning needs and to tailor their instruction accordingly. Many years later, Loris Malaguzzi, founder of the Reggio Emilia approach to early childhood education, emphasized the importance of collaboration and dialogue in learning needs analysis [12]. He believed that teachers should collaborate closely with parents, other educators, and the learners themselves to find their strengths, interests, and areas of challenge. Malaguzzi also emphasized the importance of creating a learning environment that is responsive to the social and cultural context of the learners and values creativity and exploration.

Contemporary Italian pedagogists, much like their predecessors, believe that learning needs analysis is still a critical aspect of effective education. They recognize that every learner has unique needs, interests, and abilities, and that a one-size-fits-all approach to teaching is not effective. Instead, they advocate for a personalized approach that considers each learner's individual circumstances.

The importance of a holistic approach to learning needs analysis has been emphasized [13]. Learners' needs are not limited to academic or cognitive aspects, but also include emotional, social, and spiritual dimensions. Educators should be encouraged to create a safe and supportive learning environment that addresses all these dimensions and promotes the development of the whole person. In addition to a focus on the whole person, contemporary pedagogy also emphasizes the importance of collaboration in the learning needs analysis process [14]. It recognizes that learners are not isolated individuals, but rather members of a larger community, and that the educational process should reflect this. Teachers should collaborate closely with parents, other educators, and community members to identify the needs of learners and create a learning environment that is responsive to their unique circumstances. Overall, learning needs analysis is a critical aspect of effective education, since every learner is unique, and a personalized approach that takes into account the whole person is essential: a learner-centered approach that values collaboration, creativity, and play and creates a safe and supportive learning environment for all learners. Overall, educators from all over the world have emphasized the importance of a learner-centered, holistic, and collaborative approach to learning needs analysis in higher education, industries, and professional training [15–18]. They have recognized the importance of tailoring training and education to the specific needs and goals of individual learners, and of considering the broader social, cultural, and ethical dimensions of learning.

They have also emphasized the importance of promoting critical thinking, creativity, and social responsibility, in addition to technical skills. Finally, they have emphasized the importance of collaboration between employers, educators, and learners in the learning needs analysis process in order to develop training programs that are relevant, practical, and adaptable to the changing needs of industries.

Needs analysis is a critical first step in designing an effective training intervention plan. Conducting a thorough needs analysis helps trainers to identify the specific knowledge, skills, and abilities that learners need to acquire to improve their job performance. By taking a systematic and data-driven approach to needs analysis, trainers can ensure that the training program is tailored to meet the needs of the target audience and is aligned with the goals of the organization.

The ADDIE model is a widely used instructional design framework that can be used to guide the needs analysis process [19]. The ADDIE model stands for Analysis, Design, Development, Implementation, and Evaluation. The Analysis phase of the ADDIE model is when needs analysis takes place. During this phase, trainers gather data about the current job performance of the target audience, the specific tasks and responsibilities that the training will address, and the organizational goals that the training program is intended to support.

The importance of conducting a needs analysis as part of the Analysis phase of the ADDIE model cannot be overstated. Needs analysis helps trainers to identify the specific knowledge, skills, and abilities that learners need to obtain to improve their job performance [20]. It also helps trainers to determine the right instructional methods, materials, and delivery methods that will best meet the needs of the target audience. Without a thorough needs analysis, trainers may create a training program that is not effective in addressing the specific needs of the target audience or that does not align with the goals of the organization.

## 3. The Investigation Activities

### 3.1. The Project and Themes

The activities described in this paper are related to the ERASMUS+ project entitled "Standardization and Digitalization of Rescue Education in Mining" (project acronym: DigiRescueMe). The project partners are from Turkey, Poland, Portugal, and Italy. DigiRescueMe will develop several innovative products to increase the knowledge and level of awareness of miners, rescue team members, and mining engineers and, consequently, decrease the death rate in mine accidents caused by unsafe or slow rescue processes.

In this project, semi-structured interviews and surveys were implemented, and the collected data were analyzed to identify the needs and translate them into learning requirements in terms of specific topics to address and approaches and methodologies to adopt.

Among the goals of the project was planning a curriculum, including the aims, content, processes, and scenarios, which were all identified by an initial exploratory investigation. In fact, courses will be designed to supply an answer to the learning needs identified herein.

The mining industry is a wider domain than most imagine. This sector may be linked, directly or indirectly over longer periods, to other sectors like universities, vocational schools, rescue centers, and agencies, to mention just the most probable.

For this reason, the investigations carried out herein engaged people from all these sectors to identify firstly the themes, secondly the topics, and finally the knowledge levels corresponding to those themes and topics.

In order to confirm what is written in the referenced literature, interviews were conducted during the writing of the project proposal, and focus groups were formed when the project started by engaging experts from mining companies (in Turkey and Poland); universities (in Turkey, Poland, Portugal, and Italy); and schools (in Turkey). In particular, the main findings were related to: the importance of risk management in the mining industry and methodologies and tools for identifying, assessing, and managing risks [21,22];

the critical role of mine rescue teams in responding to emergencies in the mining industry and the challenges and best practices associated with mine rescue operations [23]; and the challenges faced by workers in the mining industry and the impact of these challenges on mental health and wellbeing [24,25].

Thus, based on these studies, three themes were considered:

- Risk assessment;
- Mine rescue;
- Mental wellbeing.

Afterwards, experts identified aspects, concepts, and procedures related to these themes. Using this information, the items identified within the three topics were shared by the team of experts and, through online meetings and email exchanges, they were reviewed, cleaned up in order to avoid typos or duplications, and organized in a presentable form to facilitate investigations including surveys and the analysis of people's real needs.

Three questionnaires were prepared and delivered by COFACTOR, a Google-based system (Google Forms and Sheets) able to administer questionnaires and automatically collect and analyze data that was specifically conceived for learning needs analysis [26].

The first section of the questionnaires aimed at collecting general information on participants in the survey including: email address, country, region/city, sex, education level, current school/university/enterprise, occupation, and length of service.

The second section of each questionnaire had specific items, for which the participants had to declare, on a five-level scale, their own confidence (i.e., how much they knew about that topic).

As shown in Table 1, the questionnaire on Risk assessment had 43 items classified into three categories.

**Table 1.** Categories and number of items for Risk assessment.

| Category | Number of Items |
|---|---|
| Terminology and definitions | 21 |
| Risk management | 11 |
| Risk analyses and methods | 11 |

As shown in Table 2, the questionnaire on Mine rescue had 73 items classified into ten categories.

**Table 2.** Categories and number of items for Mine rescue.

| Category | Number of Items |
|---|---|
| Basic information | 5 |
| Formal law aspects of rescuing | 2 |
| Structure of rescue | 8 |
| Sorts and organization of rescue activities | 4 |
| Equipment used in rescue | 6 |
| Hazard classification | 15 |
| Natural hazards | 18 |
| Ergonomic, organizational, and human factor risks | 4 |
| Technical hazards | 2 |
| Methods of prevention | 9 |

As shown in Table 3, the questionnaire on Mental wellbeing had 44 items classified into five categories.

**Table 3.** Categories and number of items for Mental wellbeing.

| Category | Number of Items |
|---|---|
| Basic information | 2 |
| Risks for work-related stress | 10 |
| Stress reactions | 6 |
| Stress factors | 16 |
| Sentinel events | 10 |

### 3.2. The Participants

These investigation activities engaged an overall number of 281 participants distributed among the three surveys, mainly from Turkey and Poland. For these surveys, there was no probabilistic sampling, but they engaged all the participants from the mine companies, universities, and schools who gave their consent to take part. Details regarding the participants are collected in Tables 4–6.

**Table 4.** Number and sex of participants.

| Sex | Risk Assessment | Mine Rescue | Mental Wellbeing |
|---|---|---|---|
| Man | 144 | 163 | 116 |
| Woman | 28 | 58 | 66 |
| Not declared | 4 | 3 | 3 |
| **TOTAL** | **176** | **248** | **185** |

**Table 5.** Country of participants.

| Country | Risk Assessment | Mine Rescue | Mental Wellbeing |
|---|---|---|---|
| Poland | 35 | 44 | 48 |
| Turkey | 135 | 194 | 132 |
| Other country | 2 | 7 | 2 |
| Not declared | 4 | 3 | 3 |
| **TOTAL** | **176** | **248** | **185** |

**Table 6.** Declared education level of participants.

| Education Level | Risk Assessment | Mine Rescue | Mental Wellbeing |
|---|---|---|---|
| PhD | 2 | 2 | 1 |
| Master's degree | 46 | 61 | 47 |
| Bachelor's degree | 6 | 4 | 31 |
| High school | 86 | 118 | 64 |
| Junior high school | 18 | 37 | 18 |
| Primary school | 3 | 8 | 9 |
| Not declared | 15 | 18 | 15 |
| **TOTAL** | **176** | **248** | **185** |

Regarding the participants, to relate their identified learning needs to their declared educational level, institution, country, and role, different clusters were defined, as shown in Tables 7 and 8.

**Table 7.** Clusters of participants based on institution and country.

|  | Risk Assessment | | | Mine Rescue | | | Mental Wellbeing | | |
|---|---|---|---|---|---|---|---|---|---|
|  | Poland | Turkey | TOT | Poland | Turkey | TOT | Poland | Turkey | TOT |
| University | 9 | 36 | 45 | 7 | 53 | 60 | 8 | 33 | 41 |
| Enterprise | 18 | 64 | 82 | 21 | 94 | 115 | 18 | 62 | 80 |
| School | 8 | 28 | 36 | 14 | 43 | 57 | 19 | 24 | 43 |
| Other | 0 | 1 | 1 | 1 | 2 | 3 | 2 | 7 | 9 |
| EMPTY |  | 12 | 12 | 1 | 12 | 13 | 1 | 11 | 12 |
| TOT | 35 | 141 | 176 | 44 | 204 | 248 | 48 | 137 | 185 |

**Table 8.** Clusters of participants based on role.

|  | Risk Assessment | | | Mine Rescue | | | Mental Wellbeing | | |
|---|---|---|---|---|---|---|---|---|---|
|  | Poland | Turkey | TOT | Poland | Turkey | TOT | Poland | Turkey | TOT |
| Technical worker | 16 | 65 | 81 | 16 | 90 | 106 | 10 | 60 | 70 |
| Rescue member | 0 | 5 | 5 | 2 | 8 | 10 | 9 | 5 | 14 |
| Supervisor | 5 | 0 | 5 | 6 | 0 | 6 | 4 | 0 | 4 |
| Engineer | 3 | 7 | 10 | 6 | 9 | 15 | 4 | 5 | 9 |
| Student | 7 | 30 | 37 | 11 | 38 | 49 | 16 | 27 | 43 |
| University student | 4 | 22 | 26 | 3 | 41 | 44 | 4 | 24 | 28 |
| Other | 0 | 2 | 2 | 0 | 2 | 2 | 0 | 4 | 4 |
| EMPTY | 0 | 10 | 10 | 0 | 16 | 16 | 1 | 12 | 13 |
| TOT | 35 | 141 | 176 | 44 | 204 | 248 | 48 | 137 | 185 |

*3.3. The Collected Data*

For each topic in the questionnaires, there was an item with answers in the range of 1 to 5. A value of 1 means no knowledge about that topic (0%), and a value of 5 means full knowledge about that topic (100%). The three questionnaires are presented in the Appendices A–C.

For the 43 items on Risk assessment, the Cronbach's alpha was calculated. Its value, 0.98, meant that this questionnaire had internal consistency. Figure 1 shows the average knowledge declared on Risk assessment by the participants regarding the three identified categories of the addressed topics. Figure 2 shows the average knowledge declared by the participants on each item of Risk assessment.

For the 73 items on Mine rescue, the Cronbach's alpha was calculated. Its value, 0.99, meant that this questionnaire had internal consistency. The diagram in the Figure 3 shows the average knowledge declared regarding Mine rescue by the participants in terms of the 10 identified categories of the addressed topics. Figure 4 shows the average knowledge declared by the participants regarding each item of Mine rescue.

For the 44 items on Mental wellbeing, the Cronbach's alpha was calculated. Its value, 0.97, meant that this questionnaire had internal consistency. The diagram in the Figure 5 shows the average knowledge declared on Mental wellbeing by the participants in terms of the five identified dimension categories of the addressed topics. Figure 6 shows the average knowledge declared by the participants on each item of Mental wellbeing.

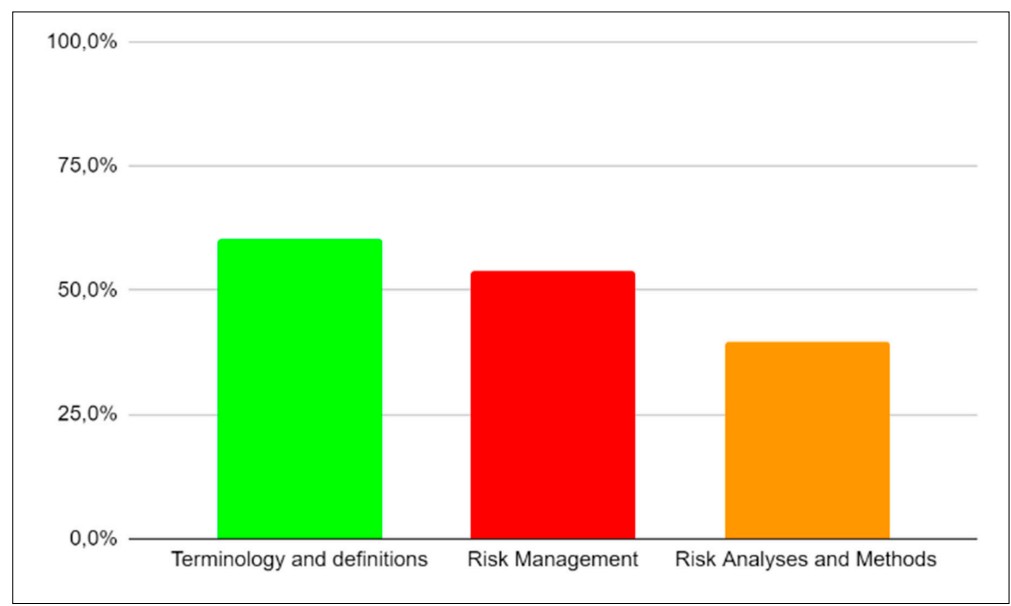

**Figure 1.** Knowledge level of participants regarding the categories of Risk assessment.

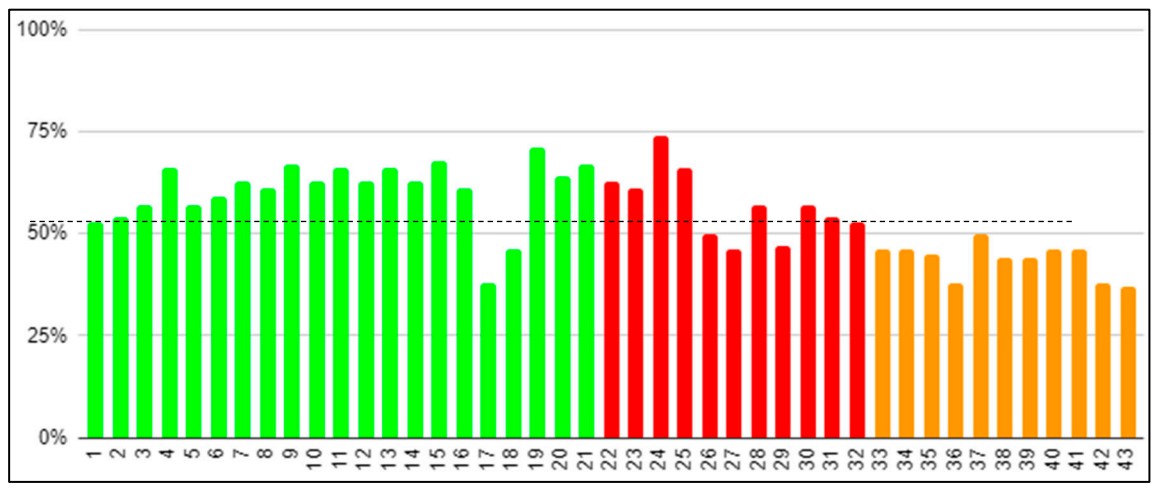

**Figure 2.** Knowledge level of participants regarding the items of Risk assessment.

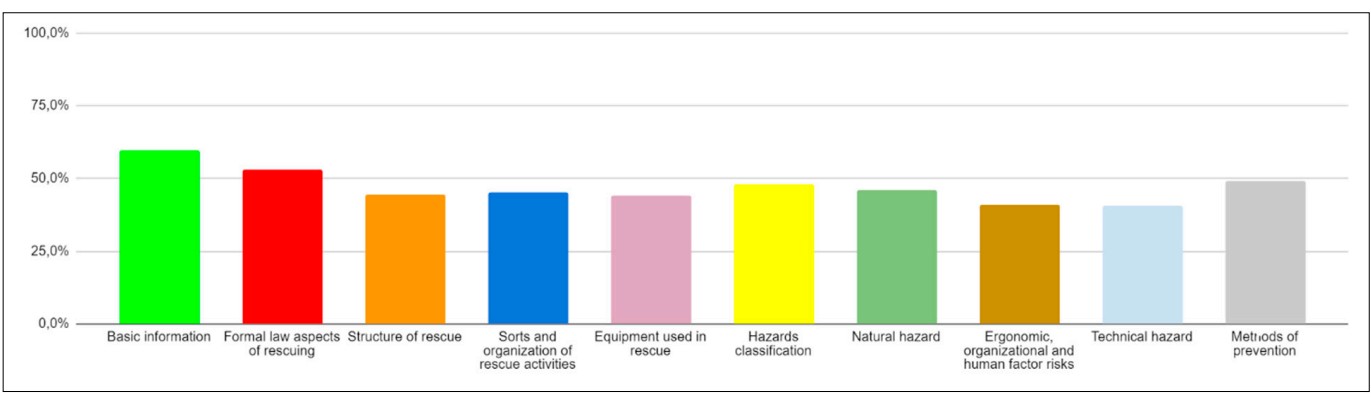

**Figure 3.** Knowledge level of participants regarding the categories of Mine rescue.

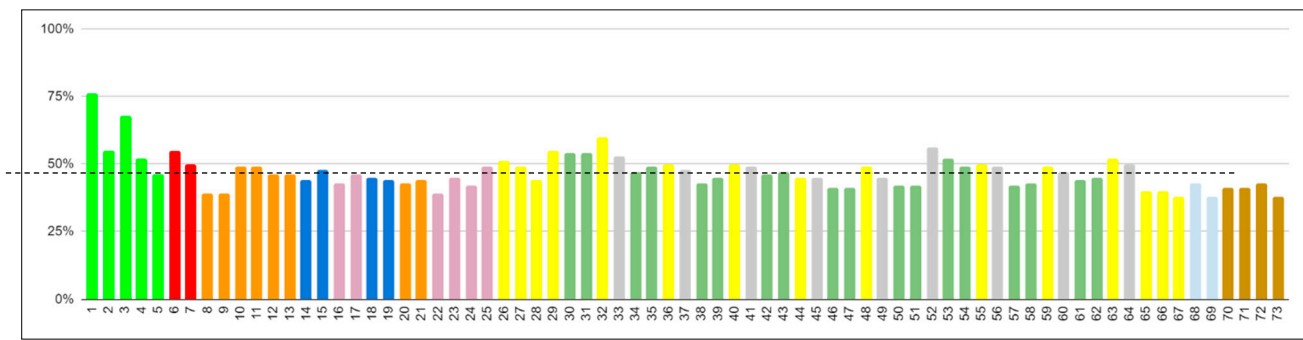

**Figure 4.** Knowledge level of participants regarding the items of Mine rescue.

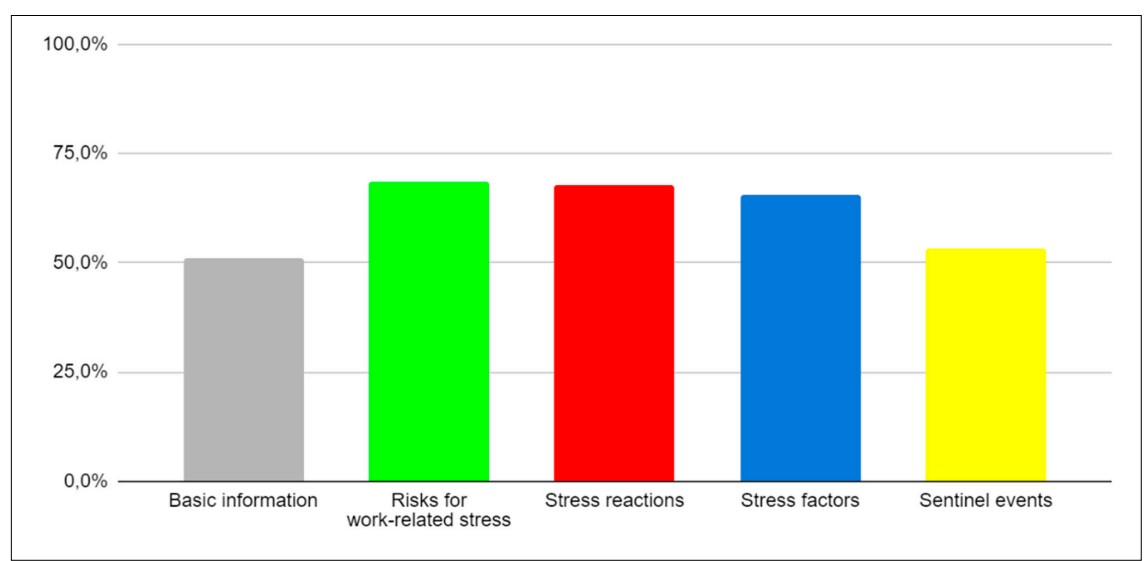

**Figure 5.** Knowledge level of participants regarding the categories of Mental wellbeing.

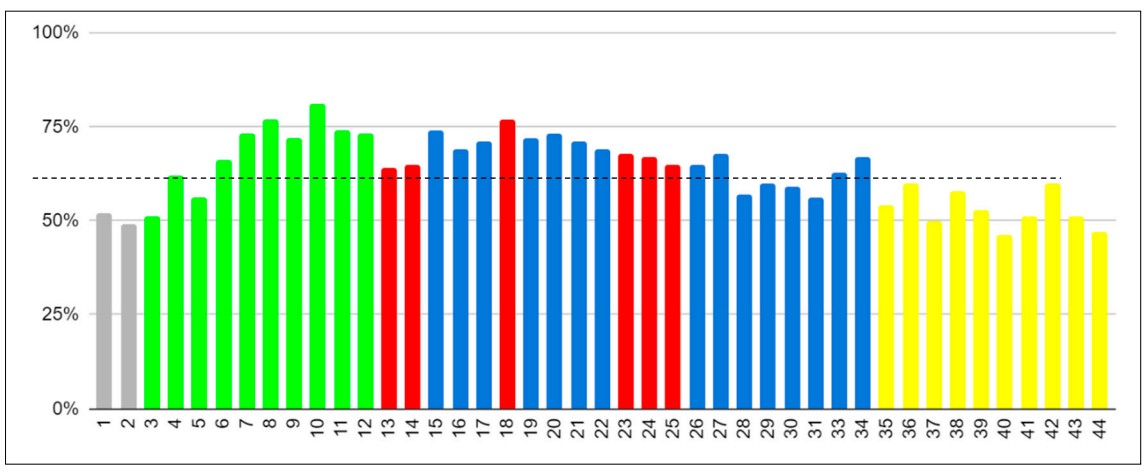

**Figure 6.** Knowledge level of participants regarding the items of Mental wellbeing.

## 4. Data analysis and Discussion

### 4.1. Data Analysis of Categories

The data collected on the categories were aggregated by considering all details declared by the participants. In this way, it was possible to distinguish them based on their country, sex, education level, institution, and role. These data are shown for each theme. Table 9

shows the data related to Risk assessment, Table 10 shows the data related to Mine rescue, and Table 11 shows the data related to Mental wellbeing.

**Table 9.** Knowledge declared by participants on categories of Risk assessment aggregated by country, sex, education level, institution, and role according to a 5-level scale (from 1 (low level of knowledge) to 5 (high level of knowledge)).

| Risk Assessment | Terminology and Definitions | Risk Management | Risk Analyses and Methods |
|---|---|---|---|
| **All (176 participants)** | 3.9 | 3.7 | 3.1 |
| **Country** | | | |
| Poland | 3.9 | 3.5 | 2.8 |
| Turkey | 3.9 | 3.8 | 3.2 |
| **Sex** | | | |
| Men | 3.9 | 3.6 | 3.0 |
| Women | 4.1 | 4.0 | 3.5 |
| **Education level** | | | |
| PhD | 4.5 | 4.2 | 3.9 |
| Master's degree | 4.0 | 3.7 | 3.0 |
| Bachelor's degree | 4.1 | 3.7 | 3.1 |
| High school | 3.9 | 3.7 | 3.1 |
| Junior high school | 3.5 | 3.4 | 2.8 |
| Primary school | 4.5 | 4.2 | 3.2 |
| **Institution** | | | |
| University | 4.1 | 3.9 | 3.3 |
| Enterprise | 3.9 | 3.6 | 2.8 |
| School | 3.7 | 3.6 | 3.3 |
| **Role** | | | |
| Technical worker | 3.9 | 3.6 | 2.8 |
| Rescue member | 4.4 | 4.5 | 4.1 |
| Supervisor | 3.9 | 3.4 | 2.8 |
| Engineer | 4.4 | 4.2 | 3.8 |
| School student | 3.5 | 3.4 | 3.1 |
| University student | 4.0 | 3.8 | 3.3 |

Regarding the data in Table 9, it is possible to note that in the "terminology and definitions" category, there were no differences between the two countries and minor differences based on sex (0.2). Wider differences could be found based on educational level (0.6), institution (0.4), and role (0.9), which could lead to different learning needs. In both the "risk management" and "risk analyses and methods" categories, there were differences between the two countries, sexes, educational levels, institutions, and roles. This meant that based on these aspects, the effective learning needs may be different and may depend on these learner's features.

Regarding the data in Table 10, it is possible to note that there were some features of the participants that may have influenced their learning needs, and others that seemed to have no effect on them. In fact, based on the country of the participants, there were no differences or slight differences (max 0.1) in the "formal law aspects of rescuing"; "ergonomic, organizational, and human factor risks"; and "technical hazards" categories. However, this feature seemed to significantly affect the other categories. Based on the

sex of the participants, there were no differences or minor differences (max 0.2) in the "basic information", "structure of rescue", "sorts and organization of rescue activities", "hazard classification", "natural hazards", and "methods of prevention" categories. However, this feature of the participants affected both the "ergonomic, organizational, and human factor risks" and "technical hazards" categories.

**Table 10.** Knowledge declared by participants on categories of Mine rescue aggregated by country, sex, education level, institution, and role according to a 5-level scale (from 1 (low level of knowledge) to 5 (high level of knowledge)).

| Mine Rescue | Basic Information | Formal Law Aspects of Rescuing | Structure of Rescue | Sorts and Organization of Rescue Activities | Equipment Used in Rescue | Hazard Classification | Natural Hazards | Ergonomic, Organizational, and Human Factor Risks | Technical Hazards | Methods of Prevention |
|---|---|---|---|---|---|---|---|---|---|---|
| **All (248 participants)** | 3.8 | 3.6 | 3.3 | 3.3 | 3.3 | 3.4 | 3.3 | 3.2 | 3.2 | 3.5 |
| **Country** | | | | | | | | | | |
| Poland | 4.4 | 3.6 | 3.8 | 3.7 | 3.5 | 3.7 | 4.1 | 3.2 | 3.2 | 3.9 |
| Turkey | 3.2 | 3.6 | 3.1 | 3.1 | 3.2 | 3.3 | 3.1 | 3.1 | 3.1 | 3.3 |
| **Sex** | | | | | | | | | | |
| Men | 3.8 | 3.6 | 3.2 | 3.3 | 3.2 | 3.4 | 3.3 | 3.1 | 3.1 | 3.4 |
| Women | 3.8 | 3.7 | 3.3 | 3.2 | 3.2 | 3.5 | 3.4 | 3.4 | 3.5 | 3.5 |
| **Education level** | | | | | | | | | | |
| PhD | 4.3 | 4.5 | 4.1 | 4.0 | 3.7 | 3.9 | 4.6 | 3.8 | 4.0 | 4.5 |
| Master's degree | 4.1 | 3.8 | 3.6 | 3.6 | 3.6 | 3.7 | 3.7 | 3.3 | 3.3 | 3.8 |
| Bachelor's degree | 4.2 | 4.3 | 3.5 | 4.0 | 3.6 | 3.7 | 3.6 | 3.5 | 3.6 | 3.6 |
| High school | 3.7 | 3.6 | 3.1 | 3.2 | 3.1 | 3.3 | 3.1 | 3.1 | 3.2 | 3.3 |
| Junior high school | 3.5 | 3.2 | 3.0 | 2.9 | 2.9 | 3.0 | 3.1 | 2.9 | 2.8 | 3.1 |
| Primary school | 4.1 | 3.6 | 3.7 | 3.7 | 3.5 | 4.0 | 3.8 | 3.5 | 3.9 | 3.8 |
| **Institution** | | | | | | | | | | |
| University | 3.9 | 3.9 | 3.3 | 3.3 | 3.3 | 3.5 | 3.4 | 3.2 | 3.2 | 3.6 |
| Enterprise | 3.8 | 3.4 | 3.2 | 3.2 | 3.2 | 3.4 | 3.2 | 3.1 | 3.0 | 3.3 |
| School | 3.8 | 3.6 | 3.3 | 3.3 | 3.3 | 3.3 | 3.4 | 3.2 | 3.2 | 3.4 |
| **Role** | | | | | | | | | | |
| Technical worker | 3.7 | 3.3 | 3.1 | 3.2 | 3.1 | 3.3 | 3.1 | 3.0 | 3.0 | 3.2 |
| Rescue member | 4.2 | 4.0 | 3.9 | 4.0 | 3.8 | 3.9 | 3.8 | 3.6 | 3.5 | 4.0 |
| Supervisor | 4.3 | 3.4 | 3.5 | 3.5 | 3.1 | 3.6 | 4.2 | 3.7 | 3.0 | 3.7 |
| Engineer | 4.5 | 4.5 | 4.3 | 4.4 | 4.2 | 4.3 | 4.4 | 4.3 | 4.2 | 4.4 |
| School student | 3.7 | 3.7 | 3.2 | 3.1 | 3.1 | 3.3 | 3.3 | 3.0 | 3.0 | 3.3 |
| University student | 3.8 | 3.9 | 2.9 | 3.1 | 3.1 | 3.6 | 3.3 | 3.5 | 3.5 | 3.8 |

The institution of the participants seemed to affect only the "methods of prevention" category and had no effect on the other categories.

Finally, based on both the education level and role of the participants, there were differences in all identified categories, and thus it was possible to relate these features of the participants to different learning needs.

Finally, regarding the data in Table 11, it is possible to note that most of the features of the participants could influence their learning needs, except the country, based on which there were no differences or slight differences (max 0.2) in all categories.

**Table 11.** Knowledge declared by participants on categories of Mental wellbeing aggregated by country, sex, education level, institution, and role according to a 5-level scale (from 1 (low level of knowledge) to 5 (high level of knowledge)).

| Mental Wellbeing | Basic Information | Risks for Work-Related Stress | Stress Reactions | Stress Factors | Sentinel Events |
|---|---|---|---|---|---|
| **All (185 participants)** | 3.6 | 4.1 | 4.2 | 4.1 | 3.7 |
| **Country** | | | | | |
| Poland | 3.6 | 4.0 | 4.1 | 4.1 | 3.6 |
| Turkey | 3.6 | 4.2 | 4.2 | 4.1 | 3.7 |
| **Sex** | | | | | |
| Men | 3.5 | 4.0 | 4.1 | 4.0 | 3.6 |
| Women | 4.2 | 4.4 | 4.4 | 4.3 | 3.8 |
| **Education level** | | | | | |
| PhD | 4.0 | 4.1 | 4.3 | 4.4 | 3.5 |
| Master's degree | 3.7 | 4.2 | 4.2 | 4.1 | 3.7 |
| Bachelor's degree | 3.6 | 4.3 | 4.4 | 4.4 | 3.8 |
| High school | 3.5 | 4.0 | 4.1 | 3.9 | 3.5 |
| Junior high school | 3.5 | 4.0 | 3.8 | 3.8 | 3.3 |
| Primary school | 4.0 | 4.4 | 4.5 | 4.5 | 4.3 |
| **Institution** | | | | | |
| University | 3.5 | 4.3 | 4.3 | 4.3 | 3.9 |
| Enterprise | 3.6 | 4.1 | 4.2 | 4.1 | 3.4 |
| School | 3.6 | 4.0 | 4.0 | 4.0 | 3.8 |
| **Role** | | | | | |
| Technical worker | 3.5 | 4.1 | 4.2 | 4.0 | 3.3 |
| Rescue member | 4.1 | 4.4 | 4.3 | 4.4 | 4.0 |
| Supervisor | 3.5 | 4.0 | 4.4 | 4.1 | 3.6 |
| Engineer | 4.0 | 4.5 | 4.6 | 4.4 | 4.0 |
| School student | 3.6 | 3.8 | 3.8 | 3.8 | 3.6 |
| University student | 3.5 | 4.3 | 4.4 | 4.4 | 4.0 |

Other features like sex, education level, institution, and role had an effect on all the categories, and thus they influenced the identified learning needs.

*4.2. Data Analysis of Items*

For all the data collected on the items, a presentation like that for the categories was chosen by considering all the details declared by the participants. For each theme, only a few items were chosen. These were the items for which the declared knowledge was less than the average value, and thus they could be seen as critical items. Moreover, statistical indexes were calculated to measure the central tendency and dispersion for each questionnaire. For the items of risk assessment, the value of the mean was 3.67, the value of the median was 3.74, the value of the mode was 4.19, and the value of the standard deviation was 1.21. For the items of mine rescue, the value of the mean was 3.40, the value of the median was 3.48, the value of the mode was 4.38, and the value of the standard deviation was 1.37. For the items of mental wellbeing, the value of the mean was 3.99, the value of the median was 4.39, the value of the mode was 4.84, and the value of the standard deviation was 1.12.

The referenced average values for the items are represented in Figures 2, 4 and 6 by dotted lines. Table 12 shows these data for risk assessment, Table 13 shows these data for mine rescue, and Table 14 shows these data for mental wellbeing.

**Table 12.** Knowledge declared by participants on the most critical items of Risk assessment aggregated by country, sex, education level, institution, and role according to a 5-level scale (from 1 (low level of knowledge) to 5 (high level of knowledge)).

| Category | Risk Analyses and Methods | | | | | |
|---|---|---|---|---|---|---|
| **Item n.** | **33** | **34** | **35** | **36** | **42** | **43** |
| **All (176 participants)** | 3.2 | 3.2 | 3.2 | 3.0 | 2.5 | 2.5 |
| **Country** | | | | | | |
| Poland | 3.1 | 3.2 | 3.1 | 3.2 | 1.8 | 2.0 |
| Turkey | 3.2 | 3.2 | 3.2 | 3.0 | 2.7 | 2.6 |
| **Sex** | | | | | | |
| Men | 3.1 | 3.1 | 3.1 | 3.0 | 2.4 | 2.4 |
| Women | 3.6 | 3.4 | 3.6 | 3.3 | 3.1 | 3.1 |
| **Education level** | | | | | | |
| PhD | 4.0 | 4.0 | 4.0 | 4.0 | 3.5 | 3.5 |
| Master's degree | 3.0 | 3.1 | 3.3 | 3.1 | 2.4 | 2.2 |
| Bachelor's degree | 3.5 | 3.3 | 3.2 | 3.3 | 1.8 | 2.2 |
| High school | 3.2 | 3.2 | 3.2 | 3.1 | 2.6 | 2.6 |
| Junior high school | 3.1 | 2.9 | 2.7 | 2.4 | 2.4 | 2.6 |
| Primary school | 3.0 | 3.3 | 3.7 | 3.3 | 2.3 | 3.0 |
| **Institution** | | | | | | |
| University | 3.2 | 3.3 | 3.4 | 3.4 | 2.9 | 2.8 |
| Enterprise | 3.1 | 2.9 | 3.0 | 2.7 | 2.1 | 2.1 |
| School | 3.3 | 3.5 | 3.3 | 3.2 | 2.8 | 2.9 |
| **Role** | | | | | | |
| Technical worker | 3.0 | 2.9 | 3.0 | 2.7 | 2.0 | 2.0 |
| Rescue member | 3.6 | 4.2 | 4.4 | 3.8 | 4.0 | 3.8 |
| Supervisor | 3.0 | 3.2 | 3.0 | 3.6 | 1.4 | 1.4 |
| Engineer | 3.9 | 3.8 | 3.9 | 3.9 | 3.5 | 3.6 |
| Student | 3.1 | 3.2 | 3.0 | 3.0 | 2.8 | 2.9 |
| University student | 3.2 | 3.3 | 3.3 | 3.4 | 2.9 | 2.8 |

The most critical items of Risk assessment in Table 12 were in the "risk analyses and methods" category. Moreover, it is possible to note that most of the features of the participants could influence their learning needs, except the country, based on which there were no differences or small differences (max 0.2) in item 33 (*I can classify the risk assessment methods*), item 34 (*I can make a comparison between qualitative and quantitative risk assessment methods*), *item 35 (*I can detect the right risk assessment method for my workplace*), *and item 36 (*I know which members are included in a risk assessment team*). *This meant that, for these items, the effective learning needs seemed to be independent of this feature of* the participant.

The most critical items of *mine rescue* in Table 13 were in the following categories: "structure of rescue"; "equipment used in rescue"; "hazard classification"; "technical hazards"; and "ergonomic, organizational, and human factor risks".

**Table 13.** Knowledge declared by participants on the most critical items of Mine rescue aggregated by country, sex, education level, institution, and role according to a 5-level scale (from 1 (low level of knowledge) to 5 (high level of knowledge)).

| Category | Structure of Rescue | Equipment Used in Rescue | Hazard Classification | Technical Hazards | Ergonomic, Organizational, and Human Factor Risks |
|---|---|---|---|---|---|
| **Item n.** | **9** | **22** | **67** | **69** | **73** |
| **All (248 participants)** | 3.0 | 3.0 | 3.1 | 3.1 | 3.1 |
| **Country** | | | | | |
| Poland | 3.9 | 3.2 | 2.8 | 2.9 | 3.1 |
| Turkey | 2.8 | 3.0 | 3.1 | 3.1 | 3.0 |
| **Sex** | | | | | |
| Men | 3.1 | 3.0 | 3.0 | 3.0 | 3.0 |
| Women | 3.0 | 3.1 | 3.3 | 3.4 | 3.3 |
| **Education level** | | | | | |
| PhD | 4.5 | 3.5 | 3.5 | 3.5 | 2.5 |
| Master's degree | 3.4 | 3.3 | 3.0 | 3.0 | 3.1 |
| Bachelor's degree | 3.0 | 3.2 | 3.7 | 3.5 | 3.2 |
| High school | 2.8 | 2.9 | 3.1 | 3.1 | 3.1 |
| Junior high school | 3.0 | 2.8 | 2.8 | 2.7 | 2.8 |
| Primary school | 3.5 | 3.3 | 3.7 | 3.8 | 3.8 |
| **Institution** | | | | | |
| University | 3.0 | 3.0 | 3.2 | 3.1 | 3.0 |
| Enterprise | 2.9 | 2.9 | 3.0 | 2.9 | 3.1 |
| School | 3.2 | 3.3 | 3.2 | 3.2 | 3.1 |
| **Role** | | | | | |
| Technical worker | 2.8 | 2.9 | 3.0 | 2.9 | 3.0 |
| Rescue member | 3.4 | 3.5 | 3.5 | 3.3 | 3.6 |
| Supervisor | 3.6 | 2.6 | 2.3 | 2.6 | 3.5 |
| Engineer | 4.4 | 4.0 | 4.1 | 4.0 | 4.4 |
| Student | 3.0 | 3.0 | 2.9 | 3.0 | 2.8 |
| University student | 2.5 | 2.5 | 3.5 | 3.4 | 3.6 |

Some features of the participants had no effect on some items. In particular, the country did not affect or had a small influence on item 22 (*I can identify the equipment used in mine rescue depending on the hazard that occurs in a particular rescue operation*), item 69 (*I can identify the most important technical hazards occurring in surface mining due to workstations*) and item 73 (*I can identify the most important organizational hazards that occur at mining workplaces*). Instead, it affected other items. The sex of the participant did not affect either item 9 (*I know who is responsible for the condition of mine rescue at a mining plant*) or item 22 (*I can identify the equipment used in mine rescue depending on the hazard that occurs in a particular rescue operation*). However, this feature had an influence on other identified items. The institution of the participant had no effect or a small influence on both item 67 (*I know the prevention of the most important hazards occurring in surface mining*) and item 73 (*I can identify the most important organizational hazards that occur at mining workplaces*). How-

ever, it affected other items. Finally, other features of the participants affected these items, and thus they had an influence on the detected learning needs.

**Table 14.** Knowledge declared by participants on the most critical items of Mental wellbeing aggregated by country, sex, education level, institution, and role according to a 5-level scale (from 1 (low level of knowledge) to 5 (high level of knowledge)).

| Category | Risks for Work-Related Stress | Sentinel Events | | | |
|---|---|---|---|---|---|
| Item n. | 3 | 36 | 39 | 42 | 43 |
| **All (185 participants)** | 3.5 | 3.5 | 3.4 | 3.5 | 3.4 |
| **Country** | | | | | |
| Poland | 3.3 | 3.0 | 3.3 | 3.6 | 3.5 |
| Turkey | 3.6 | 3.7 | 3.5 | 3.6 | 3.4 |
| **Sex** | | | | | |
| Men | 3.4 | 3.4 | 3.4 | 3.5 | 3.4 |
| Women | 4.0 | 3.7 | 3.7 | 3.8 | 3.4 |
| **Education level** | | | | | |
| PhD | 4.0 | 2.0 | 4.0 | 4.0 | 4.0 |
| Master's degree | 3.6 | 3.5 | 3.5 | 3.4 | 3.6 |
| Bachelor's degree | 3.6 | 3.8 | 3.6 | 3.6 | 3.8 |
| High school | 3.3 | 3.3 | 3.3 | 3.6 | 3.1 |
| Junior high school | 3.1 | 3.2 | 3.1 | 3.3 | 3.0 |
| Primary school | 3.8 | 4.2 | 3.8 | 4.4 | 4.3 |
| **Institution** | | | | | |
| University | 3.5 | 3.9 | 3.8 | 3.8 | 3.7 |
| Enterprise | 3.3 | 3.4 | 3.1 | 3.2 | 3.1 |
| School | 3.8 | 3.3 | 3.6 | 3.8 | 3.7 |
| **Role** | | | | | |
| Technical worker | 3.4 | 3.2 | 3.1 | 3.2 | 3.0 |
| Rescue member | 4.0 | 4.0 | 3.5 | 3.7 | 4.0 |
| Supervisor | 2.7 | 3.0 | 3.2 | 3.7 | 3.7 |
| Engineer | 4.3 | 3.7 | 4.1 | 4.0 | 4.1 |
| Student | 3.5 | 3.3 | 3.5 | 3.6 | 3.5 |
| University student | 3.3 | 4.0 | 4.1 | 3.8 | 3.8 |

The most critical items of Mental wellbeing in Table 14 were in two categories: "risks for work-related stress" and "sentinel events". Among these items, only item 43 (*The malaise of a worker results from his/her low job satisfaction*) showed no differences based on sex and slight differences based on country (0.1) and institution (0.1). For other items, there were differences based on the features of the participants, which meant that they were effectively related to their own learning needs.

## 5. Discussion

The world of the mining industry, however ancient, is still a context to which multiple production activities are anchored. For this reason, the set of professional figures and their cultural backgrounds and education levels are also very varied. There are mining companies that have needs and problems of many types, on which the specific training

activities of schools and universities converge and for which the innovation and technology transfer activities of universities and research centers are fundamental.

However, this is a dangerous area, and therefore it is a fertile ground for initiatives that have the aim of improving the safety and health of those who work in the industry. For the same reason, training activities are also welcomed when they are related to hazards, risks, rescues, and the delicate objective of saving human lives. The recurring problem, however, is to make these activities more effective and efficient so as to reconcile them with everyday work and life and maximize results by minimizing their costs. It is known that the analysis of needs is the first fundamental step to achieving this goal. Unfortunately, for this very varied area, there are no specific studies that have been conducted to investigate and analyze the training needs.

The DigiRescueMe project was born precisely for this reason: to intervene on issues and problems pertinent to the mining industry by preparing targeted training courses.

In this investigation, a large amount of data was collected. Only a few of them were presented and analyzed herein with the aim of identifying learning gaps among the participants regarding mine rescue, risk assessment, and mental wellbeing. Figure 7 represents these gaps in order to clarify where the learning needs lie and lead the future development of customized and more effective learning courses for particular learner profiles [27].

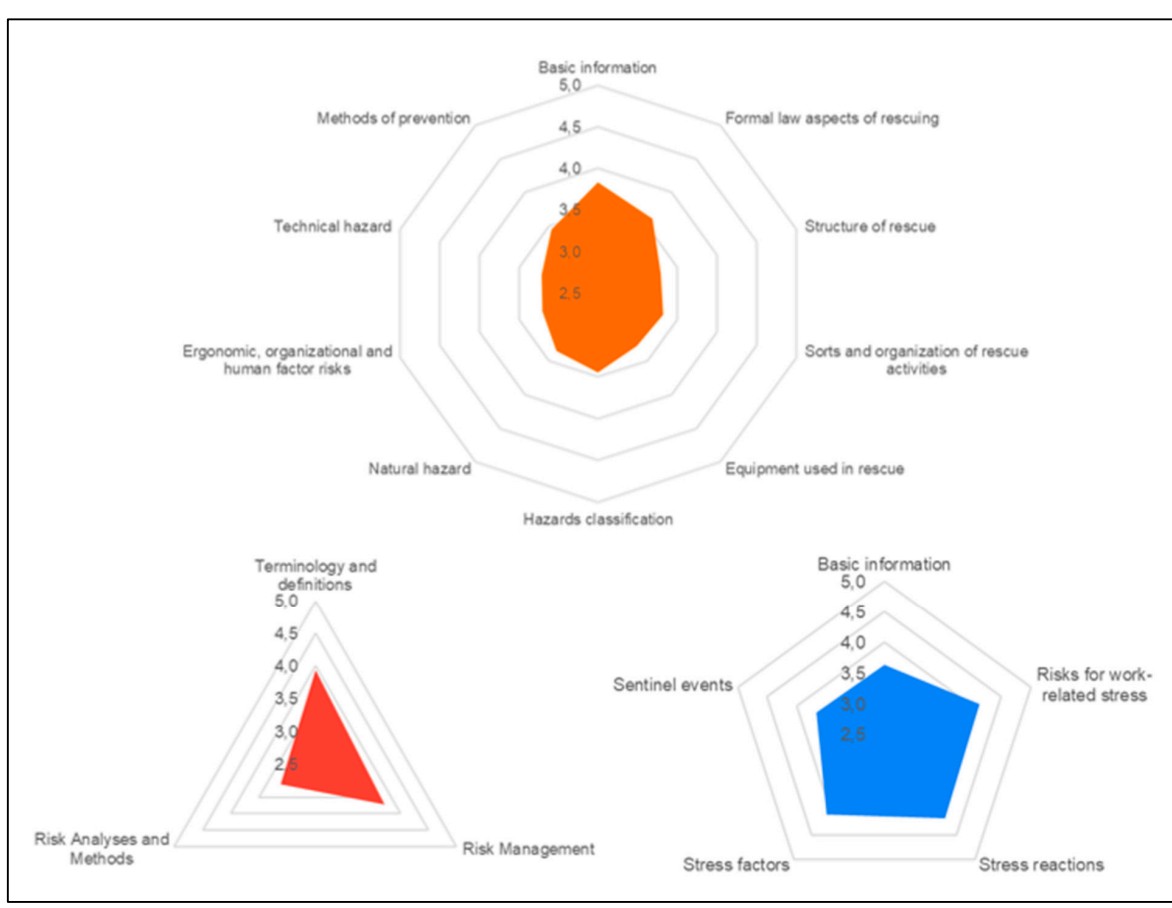

**Figure 7.** Learning gaps among participants regarding Mine rescue, Risk assessment, and Mental wellbeing according to a 5-level scale (from 1 (low level of knowledge) to 5 (high level of knowledge)).

## 6. Conclusions

Learning needs analysis is a process that helps identify the specific learning needs of adult learners. This process is crucial to create effective adult education courses, as it ensures that the content and approach are tailored to the learners' specific needs and goals.

Adult learners have different learning needs than children, as their motivations and goals for learning are often different. Adults are typically motivated by practical goals, such as advancing their careers or gaining new skills to enrich their personal lives. They also tend to have more life experience and knowledge than children, which can affect their learning styles and preferences [28]. To effectively meet the learning needs of adult learners, instructors must understand: learners' prior knowledge and experiences, their goals and motivations for learning, their preferred learning styles and approaches, their level of engagement and participation, and finally the challenges they face and obstacles to learning they must overcome. By understanding these factors, instructors can design courses that are relevant, engaging, and effective for adult learners [29].

Thus, this kind of analysis is a critical step in creating effective courses for adult learners. In fact, the activities described in this paper may lead to tailored content, improved learning outcomes, the increased engagement of participants, and reduced costs.

The learning needs analysis conducted in this project will help instructors to tailor course content to the specific needs of the adult learners in terms of the identified themes and topics related to Risk assessment, Mine rescue, and Mental wellbeing. By identifying the learners' goals, preferences, and prior knowledge, instructors can ensure that the content is relevant and engaging for all contexts (countries and institutions), all the abovementioned education levels, and all the roles the learners actually have or aim to have [30].

What emerged from the learning needs analysis was that the training interventions will be prepared in the three considered macro-areas with the aim of increasing knowledge, skills, and attitudes on not only a strictly cognitive and technical-scientific level, but, above all, a socio-psychological level. What has been stated translates into a plan for skills where strategies and techniques of a socio-constructivist nature will be adopted, paying particular attention to the active role of the learners in their learning processes. The teaching interventions will be planned by proposing real situations, enhancing the student's concrete experience in real or simulated contexts. For this reason, theoretical interventions will be followed by laboratory activities that will require, for example, the integration of content from different disciplines in the analysis and resolution of real cases through the adoption of methods to critically interpret the observed phenomena.

These preliminary studies ensure that both the course content and the approach that will be developed in the DigiRescueMe project will be aligned with the learners' goals and preferences, leading to improved learning outcomes. In this way, learners will feel that the course content is relevant to their needs and interests, so they will be more likely to be engaged and motivated to learn. This approach will surely increase engagement and participation. Learners will not feel that they are wasting time, and the involved institutions will be sure that they are investing resources and money in the right way by seeing that the courses are designed effectively and efficiently for the people working and or studying there.

In conclusion, learning needs analysis is a critical process for creating effective courses for adult learners. By understanding the learners' goals, preferences, and prior knowledge, instructors can design courses that are relevant, engaging, and effective. The activities described in this paper were conducted with the aim of leading the DigiRescueMe project to improved learning outcomes, increased engagement, and cost savings. At the moment, the obtained results provide a very convincing basis, and it is hoped that they will be useful for the future development of the project and the activities that can be generated by it.

**Author Contributions:** Conceptualization, S.M. and A.M.; Methodology, S.M. and R.V.; Formal analysis, S.M.; Investigation, R.V.; Supervision, A.M. All authors have read and agreed to the published version of the manuscript.

**Funding:** This research has been done in the project 2021-1-TR01-KA220-VET-000028090 named DigiRescueMe, Standardization and Digitalization of Rescue Education in Mining funded by the ERASMUS+ programme of the European Union.

**Institutional Review Board Statement:** Social and Human Sciences Scientific Research and Publication Ethics Committee of Kütahya Dumlupınar University (protocol code 147851 and date of approval: 17 October 2022).

**Informed Consent Statement:** Informed consent was obtained from all subjects involved in the study.

**Data Availability Statement:** All information related to the cited project may be founded at http://digirescueme.com.

**Acknowledgments:** These studies have been employed for the need analysis in the work package *O1 – Development of Standardised Rescue Curriculum* – of the project 2021-1-TR01-KA220-VET-000028090 named *DigiRescueMe, Standardization and Digitalization of Rescue Education in Mining*, funded by the ERASMUS+ programme of the European Union. For the administration of the questionnaires, the interviews and many fruitful discussions, special thanks are addressed to colleagues from the partner institutions: *Kutahya Dumlupinar Universitesi*, Turkey (DPU), *Akademia Gorniczo-Hutnicza Im. Stanislawa Staszica W Krakowie*, Poland (AGH), *Universidade Do Porto*, Portugal (Uporto) and *Nurettin Çarmıklı Madencilik Mesleki ve Teknik Anadolu Lisesi*, Turkey (BalVET).

**Conflicts of Interest:** The authors declare no conflict of interest.

**Appendix A. Questionnaire on Risk Assessment**

**Dimensions**

1. Terminology and definitions;
2. Risk management;
3. Risk analyses and methods.

**Answers**

1. Not at all;
2. A little;
3. So and so;
4. Enough;
5. A lot.

**Demographic questions**

1. Country (Where you actually work/study);
2. Region/City (Where you actually work/study);
3. Gender (M, F);
4. Education level (The highest study title you have (Diploma, Degree, Master, PhD, . . .));
5. School, University, Enterprise, Mine (Where you actually work/study);
6. Occupation (Rescue member, Teacher, Student, Miner, Engineer, . . .);
7. Length of service (How many years (for the workers)).

**Items**

| | About Risk Assessment. . . | Dimension |
|---|---|---|
| 1 | I can give a definition of the hazard at the workplace. | Terminology and definitions |
| 2 | I can identify various hazardous situations in my workplace. | Terminology and definitions |
| 3 | I can determine prevention measures that I should take into account related to hazard. | Terminology and definitions |
| 4 | After the identification of the hazardous situation, I know steps which I should follow. | Terminology and definitions |
| 5 | I can distinguish physical hazards from chemical hazards. | Terminology and definitions |
| 6 | I know the difference between hazard and risk. | Terminology and definitions |
| 7 | I can define what a risk is at the workplace. | Terminology and definitions |
| 8 | I can determine risky situations at my workplace. | Terminology and definitions |

| 9 | I can define a near miss at the workplace. | Terminology and definitions |
|---|---|---|
| 10 | I can give an example of a near miss. | Terminology and definitions |
| 11 | I know how to fill the near miss form at my workplace. | Terminology and definitions |
| 12 | I can create a near miss form at my workplace. | Terminology and definitions |
| 13 | I can define the accident at the workplace. | Terminology and definitions |
| 14 | I can distinguish workplace accidents from other types of accidents. | Terminology and definitions |
| 15 | While I am making a plan, I consider the PDCA Cycle. | Terminology and definitions |
| 16 | I know the process and steps of the planning for any risk assessment process. | Terminology and definitions |
| 17 | I think that teamwork may contribute to individual work in risk assessment process. | Terminology and definitions |
| 18 | I think that individual work may contribute to teamwork in risk assessment process. | Terminology and definitions |
| 19 | I can distinguish which hazard resources are related to man. | Terminology and definitions |
| 20 | I can distinguish which hazard resources are related to machine. | Terminology and definitions |
| 21 | I can distinguish which hazard resources are related to medium. | Terminology and definitions |
| 22 | I can distinguish which hazard resources are related to mission. | Risk management |
| 23 | I know which hazards are the riskiest at the workplace. | Risk management |
| 24 | I know the control hierarchy. | Risk management |
| 25 | I know the principles of the risk assessment at the workplace. | Risk management |
| 26 | I am aware of the benefits of risk assessment for my workplace. | Risk management |
| 27 | I know how to define risk management steps at the workplace. | Risk management |
| 28 | I know when I should carry out the risk assessment at the workplace. | Risk management |
| 29 | I know when I should repeat the risk assessment at the workplace. | Risk management |
| 30 | I know the revision process of risk assessment at the workplace. | Risk management |
| 31 | I know the international legislation about occupational health and safety at the workplace. | Risk management |
| 32 | I know the national legislation about occupational health and safety at the workplace. | Risk management |
| 33 | I can classify the risk assessment methods. | Risk analyses and methods |
| 34 | I can make a comparison between qualitative and quantitative risk assessment methods. | Risk analyses and methods |
| 35 | I can detect the right risk assessment method for my workplace. | Risk analyses and methods |
| 36 | I know which members are included in a risk assessment team. | Risk analyses and methods |
| 37 | I know the roles of the members in a risk assessment team. | Risk analyses and methods |
| 38 | I can determine the dangerous risks at the workplace. | Risk analyses and methods |
| 39 | I can sort of the risks from the most dangerous one to least one. | Risk analyses and methods |
| 40 | I know how to implement Fine-Kinney Method. | Risk analyses and methods |
| 41 | I know how to implement John-Ridley Method. | Risk analyses and methods |
| 42 | I know how to implement HAZOP Method. | Risk analyses and methods |
| 43 | I know how to implement What If Method. | Risk analyses and methods |

## Appendix B. Questionnaire on Mine Rescue

**Dimensions**

1. Basic information;
2. Formal law aspects of rescuing;
3. Structure of rescue;

4. Sorts and organization of rescue activities;
5. Equipment used in rescue;
6. Hazard classification;
7. Natural hazards;
8. Ergonomic, organizational, and human factor risks;
9. Technical hazards;
10. Methods of prevention.

**Answers**

1. Not at all;
2. A little;
3. So and so;
4. Enough;
5. A lot.

**Demographic questions**

1. Country (Where you actually work/study);
2. Region/City (Where you actually work/study);
3. Gender (M, F);
4. Education level (The highest study title you have (Diploma, Degree, Master, PhD, ...));
5. School, University, Enterprise, Mine (Where you actually work/study);
6. Occupation (Rescue member, Teacher, Student, Miner, Engineer, ...);
7. Length of service (How many years (for the workers)).

**Items**

| | About Mine Rescue... | Dimension |
|---|---|---|
| 1 | I know what occupational health and safety is. | Basic information |
| 2 | I know what mine rescue involves. | Basic information |
| 3 | I can distinguish between a rescue team and a rescue squad. | Basic information |
| 4 | I know what natural hazards in mining are. | Basic information |
| 5 | I know what technical hazards in mining are. | Basic information |
| 6 | I know the basic legal acts on the mining activity. | Formal law aspects of rescuing |
| 7 | I know the basic legal acts on the mine rescue. | Formal law aspects of rescuing |
| 8 | I know who supervises and controls mine rescue at mining facilities. | Structure of rescue |
| 9 | I know who is responsible for the condition of mine rescue at a mining plant. | Structure of rescue |
| 10 | I know what a mine rescue plan is. | Structure of rescue |
| 11 | I know the organizational structure of mine rescue services. | Structure of rescue |
| 12 | I know the basic tasks of the services and organizations that deal with the mine rescue. | Structure of rescue |
| 13 | I know what tasks are performed by the manager of the rescue operation. | Structure of rescue |
| 14 | I know the signals used in the mine rescue. | Sorts and organization of rescue activities |
| 15 | I know what escape routes are in mines. | Sorts and organization of rescue activities |
| 16 | I can indicate the basic equipment of the rescue squad on duty. | Equipment used in rescue |
| 17 | I can identify the basic equipment of the rescue base. | Equipment used in rescue |
| 18 | I know what the preventive works performed by mine rescuers are. | Sorts and organization of rescue activities |

| 19 | I can identify the types of rescue operations carried out in mining plants. | Sorts and organization of rescue activities |
|---|---|---|
| 20 | I know how a rescue operation should be managed. | Structure of rescue |
| 21 | I know the responsibilities of mine plant managers and supervisors involved in rescue operations. | Structure of rescue |
| 22 | I can identify the equipment used in mine rescue depending on the hazard that occurs in a particular rescue operation. | Equipment used in rescue |
| 24 | I know what medical equipment a rescue team should have access to. | Equipment used in rescue |
| 25 | I know the types of fire extinguishers used in the mine rescue. | Equipment used in rescue |
| 26 | I can distinguish between the different divisions of hazards used in mining operations. | Hazard classification |
| 27 | I know the difference between hazardous, harmful and annoying factors of the work environment. | Hazard classification |
| 28 | I can make a distinction between hazards in the working environment due to the nature of the impact of the factor. | Hazard classification |
| 29 | I can identify natural hazards that occur in mining due to the nature of the hazard. | Hazard classification |
| 30 | I know what a methane hazard is. | Natural hazards |
| 31 | I know the causes of methane hazard in mines. | Natural hazards |
| 32 | I know the classification of methane hazard. | Hazard classification |
| 33 | I know the basic methods of prevention of methane hazard. | Methods of prevention |
| 34 | I know what coal dust explosion hazard is. | Natural hazards |
| 35 | I know the causes of coal dust explosion hazard. | Natural hazards |
| 36 | I know the classification of coal dust explosion hazard. | Hazard classification |
| 37 | I know the basic methods of prevention of coal dust explosion hazard. | Methods of prevention |
| 38 | I know what the danger of endogenous fires is. | Natural hazards |
| 39 | I know the reasons for the occurrence of endogenous fire danger. | Natural hazards |
| 40 | I know the classification of the danger of endogenous fires. | Hazard classification |
| 41 | I know the basic methods of prevention of endogenous fire danger. | Methods of prevention |
| 42 | I know what climatic hazard is. | Natural hazards |
| 43 | I know the causes of climatic hazard. | Natural hazards |
| 44 | I know the classification of climatic hazard. | Hazard classification |
| 45 | I know the basic methods of prevention of climatic hazard. | Methods of prevention |
| 46 | I know what radiation hazard is. | Natural hazards |
| 47 | I know the causes of the occurrence of radiation hazard. | Natural hazards |
| 48 | I know the classification of radiation hazard. | Hazard classification |
| 49 | I know the basic methods of prevention of radiation hazard. | Methods of prevention |
| 50 | I know what the danger of roof fall. | Natural hazards |
| 51 | I know the causes of the danger of roof fall. | Natural hazards |
| 52 | I know the basic methods of prevention of roof fall | Methods of prevention |
| 53 | I know what the danger of tremors and quakes. | Natural hazards |
| 54 | I know the reasons for the occurrence of tremors and quakes. | Natural hazards |
| 55 | I know the classification of tremors. | Hazard classification |
| 56 | I know the basic methods of prevention of tremors. | Methods of prevention |

| 57 | I know what the hazard of gas and rock outbursts is. | Natural hazards |
|---|---|---|
| 58 | I know the reasons for the occurrence of gas and rock outbursts hazard. | Natural hazards |
| 59 | I know the classification of the hazard of gas and rock outbursts. | Hazard classification |
| 60 | I know the basic methods of prevention of the danger of gas and rock outbursts. | Methods of prevention |
| 61 | I know what a water hazard is. | Natural hazards |
| 62 | I know the causes of water hazard. | Natural hazards |
| 63 | I know the classification of water hazard. | Hazard classification |
| 64 | I know the basic methods of water hazard prevention. | Methods of prevention |
| 65 | I know the classification of the most important hazards that are present in surface mining. | Hazard classification |
| 66 | I can identify the most important hazards occurring in surface mining. | Hazard classification |
| 67 | I know the prevention of the most important hazards occurring in surface mining. | Hazard classification |
| 68 | I can indicate the most important technical hazards occurring in underground mining due to workplaces. | Technical hazards |
| 69 | I can identify the most important technical hazards occurring in surface mining due to workstations. | Technical hazards |
| 70 | I know what ergonomic hazards are. | Ergonomic, organizational, and human factor risks |
| 71 | I can identify the most important ergonomic hazards occurring in mining workstations. | Ergonomic, organizational, and human factor risks |
| 72 | I know what organizational hazards are. | Ergonomic, organizational, and human factor risks |
| 73 | I can identify the most important organizational hazards that occur at mining workplaces. | Ergonomic, organizational, and human factor risks |

### Appendix C. Questionnaire on Mental Wellbeing

**Dimensions**

1. Basic information;
2. Risks for work-related stress;
3. Stress reactions;
4. Stress factors;
5. Sentinel events.

**Answers**

1. Strongly disagree;
2. Disagree;
3. I don't know;
4. Agree;
5. Strongly agree.

**Demographic questions**

1. Country (Where you actually work/study);
2. Region/City (Where you actually work/study);
3. Gender (M, F);
4. Education level (The highest study title you have (Diploma, Degree, Master, PhD, ...));
5. School, University, Enterprise, Mine (Where you actually work/study);
6. Occupation (Rescue member, Teacher, Student, Miner, Engineer, ...);
7. Length of service (How many years (for the workers)).

**Items**

| | About Mental Wellbeing... | Dimension |
|---|---|---|
| 1 | I can define the most important psychosocial hazards in the work environment. | Basic information |
| 2 | I know the preventive measures related to psychosocial hazards occurring in the work environment. | Basic information |
| 3 | Risks for work-related stress are related to the use of new forms of employment contracts. | Risks for work-related stress |
| 4 | Temporary contracts and the uncertainty and insecurity of the job itself is a risk for work-related stress. | Risks for work-related stress |
| 5 | An increasingly aging workforce is a risk for work-related stress. | Risks for work-related stress |
| 6 | The lack of adequate turnover is a risk for work-related stress. | Risks for work-related stress |
| 7 | High workload is a risk for work-related stress. | Risks for work-related stress |
| 8 | Pressure on workers by management is a risk for work-related stress. | Risks for work-related stress |
| 9 | High emotional tension is a risk for work-related stress. | Risks for work-related stress |
| 10 | Violence and harassment at work is a risk for work-related stress. | Risks for work-related stress |
| 11 | Interference of private life in the work is a risk for work-related stress. | Risks for work-related stress |
| 12 | Imbalance between work and private life is a risk for work-related stress. | Risks for work-related stress |
| 13 | Harmful physical reactions to job demands are signals of work-related stress. | Stress reactions |
| 14 | Harmful emotional reactions to job demands are signals of work-related stress. | Stress reactions |
| 15 | Work related stress can be raised when job demands are not commensurate with the skills, resources or needs of workers. | Stress factors |
| 16 | Work related stress can be raised when people perceive an imbalance between the demands made on them and the resources available to them to meet those demands. | Stress factors |
| 17 | Stress is a situation of prolonged tension at work. | Stress factors |
| 18 | Stress can reduce efficiency at work and lead to poor health. | Stress reactions |
| 19 | Work-related stress can be caused by the content of the job. | Stress factors |
| 20 | Work-related stress can be caused by inadequacy in the management of the work organization. | Stress factors |
| 21 | Work-related stress can be caused by inadequacy in the management of the working environment. | Stress factors |
| 22 | Work-related stress can be caused by shortcomings in communication. | Stress factors |
| 23 | Work-related stress is a condition that can be accompanied by physical disorders. | Stress reactions |
| 24 | Work-related stress is a condition that can be accompanied by psychological disorders. | Stress reactions |
| 25 | Work-related stress is a condition that can be accompanied by social dysfunctions. | Stress reactions |
| 26 | Work-related stress is a consequence of the fact that some individuals do not feel capable of responding to the requests or expectations placed on them. | Stress factors |
| 27 | The work environment is one of the factors that cause work-related stress. | Stress factors |
| 28 | The equipment is one of the factors that cause work-related stress. | Stress factors |
| 29 | The workloads and rhythms are factors that cause work-related stress. | Stress factors |
| 30 | The role of the organization is one of the factors that cause work-related stress. | Stress factors |
| 31 | The decision-making autonomy and control are one of the factors that cause work-related stress. | Stress factors |
| 32 | The legal responsibility inside an organization is one of the factors that cause stress. | Stress factors |
| 33 | Interpersonal conflicts at work are factors that cause work-related stress. | Stress factors |
| 34 | The career evolution and development are one of the factors that cause work-related stress. | Stress factors |

| 35 | Accident rates are sentinel events of work-related stress. | Sentinel events |
|----|------------------------------------------------------------|-----------------|
| 36 | The absence due to illness is one of the sentinel events of work-related stress. | Sentinel events |
| 37 | The turnover is one of the sentinel events of work-related stress. | Sentinel events |
| 38 | The number of proceedings and sanctions is one of the sentinel events of work-related stress. | Sentinel events |
| 39 | The number of reports to the competent doctor is one of the sentinel events of work-related stress. | Sentinel events |
| 40 | The specific and frequent formalized complaints by workers are sentinel events of work-related stress. | Sentinel events |
| 41 | Accident rates are sentinel events of work-related stress. | Sentinel events |
| 42 | The malaise of a worker results from his/her poor identification with the group or organization to which he/she belongs to. | Sentinel events |
| 43 | The malaise of a worker results from his/her low job satisfaction. | Sentinel events |
| 44 | The malaise of a worker results from his/her low confidence in the organization and the consideration of wanting to leave his/her job. | Sentinel events |

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
