# Peer review of "An Investigation of Learning Needs in the Mining Industry"

_education, doi:10.3390/educsci13101036_

Round 1

Reviewer 1 Report

The article highlights the importance of learning needs analysis for adult learners. In the conclusion the authors briefly mention that the results obtained will be useful for future developments of the project and related activities. It would be helpful to expand on this point by discussing the potential implications of the research findings for future courses or projects in the mining industry or adult education. Consider providing some concrete examples of how the results of the learning needs analysis could be practically applied to course design within the DigiRescueMe project. This can help readers envision the real-world impact of your work.

Author Response

Thank you very much for your suggestion!

The Section “6. Conclusions” has been integrated in order to clarify how these activities are important in general and, in particular, they are fundamental for the  success of the mentioned project.

Reviewer 2 Report

Mine safety is strongly related with the mining method applied in the mine, the mine planning and design and the engineering applied in the mining operation.

The greatest interest of this publication is to have made an important effort to define generic bases that can serve as a reference for a very wide set of mining activities. In this sense, it is appropriate to congratulate the authors for the effort made.

Given the evolution that has occurred in mining in recent years, a greater number of more recently published references to base this paper is something to be found missing. Likewise, a certain classification of them is also missing, since open pit mining operations do not have the same degree of accidents or risk as underground operations.

Mine safety is an ongoing commitment. It's not enough to provide initial training; continuous reinforcement and a proactive approach to safety are essential for a safer mining environment.

Author Response

Thank you very much!

Reviewer 3 Report

Interesting study. Detailed comments are included in the annotated version of the paper. 

Clarity is required in some portions of the paper. 

Minor edits needed as indicated in the annotated paper. 

Author Response

Thank you very much for your punctual comments.

As showed in the following table, we tried to provide answers to all of them.

Reviewer’s comments

Authors’ answer

Pag 1

The abstract has been modified

Pag.2

The sentence on row 57 has been amended.

The mistake on row 73 has been corrected.

Pag.4

The sentences starting on row 184 has been rephrased.

“Highlighted” on row 195 has been changed with “identified”.

The sentences starting on row 196 has been rephrased.

Pag.5

It has been clarified what are the items.

It has been done for all the questionnaires and related tables on the following pages.

Pag.10

The first paragraph of 4.1. Data analysis on categories has been clarified.

The caption of the table 9 has been completed to explain the numbers.

Pag.11

The caption of the table 10 has been completed to explain the numbers.

The explanation in the text has been modified.

Pag.12

The caption of the table 11 has been completed to explain the numbers.

The explanation in the text has been modified.

Pag.13

The heading of the table 12 has been modified and a caption on the second row of the table has been added.

Pag.14

The heading of the table 13 has been modified and a caption on the second row of the table has been added.

Pag.15

The heading of the table 14 has been modified and a caption on the second row of the table has been added.

Pag.16

The sentence “in some countries almost considered obsolete” has been deleted.

The end of the section has been integrated with a sentence referring the Fig.7.